# Anatomical-MRI Correlations in Adults and Children with Hypertrophic Cardiomyopathy

**DOI:** 10.3390/diagnostics12020489

**Published:** 2022-02-14

**Authors:** Radu Ovidiu Rosu, Ana Lupsor, Alexandru Necula, Gabriel Cismaru, Simona Sorana Cainap, Daniela Iacob, Cecilia Lazea, Andrei Cismaru, Alina Gabriela Negru, Dana Pop, Gabriel Gusetu

**Affiliations:** 1Fifth Department of Internal Medicine, Cardiology Rehabilitation, 400347 Cluj-Napoca, Romania; rosu.radu1053@gmail.com (R.O.R.); pop.dana@umfcluj.ro (D.P.); gusetu@gmail.com (G.G.); 2Iuliu Hatieganu University of Medicine and Pharmacy, 400012 Cluj-Napoca, Romania; alexnecula10@gmail.com (A.N.); cainap.simona@gmail.com (S.S.C.); iacobdaniela777@gmail.com (D.I.); cicilazearo@yahoo.com (C.L.); cismaru_andrei@yahoo.com (A.C.); 32nd Pediatric Department, Mother and Child Department, Emergency Clinical Hospital for Children, 400177 Cluj-Napoca, Romania; 43rd Pediatric Department, Mother and Child Department, Emergency Clinical Hospital for Children, 400217 Cluj-Napoca, Romania; 51st Pediatric Department, Mother and Child Department, Emergency Clinical Hospital for Children, 400370 Cluj-Napoca, Romania; 6Research Center for Functional Genomics, Biomedicine and Translational Medicine, 400337 Cluj-Napoca, Romania; 7Department of Cardiology, ‘Victor Babeș’ University of Medicine and Pharmacy of Timisoara, 300041 Timisoara, Romania; eivanica@yahoo.com

**Keywords:** hypertrophic cardiomyopathy, interstitial fibrosis, myocyte disarray, cardiac MRI, delayed enhancement, apical aneurysm, sudden cardiac death

## Abstract

Hypertrophic Cardiomyopathy (HCM) is the most frequent hereditary cardiovascular disease and the leading cause of sudden cardiac death in young individuals. Advancements in CMR imaging have allowed for earlier identification and more accurate prognosis of HCM. Interventions aimed at slowing or stopping the disease’s natural course may be developed in the future. CMR has been validated as a technique with high sensitivity and specificity, very few contraindications, a low risk of side effects, and is overall a good tool to be employed in the management of HCM patients. The goal of this review is to evaluate the magnetic resonance features of HCM, starting with distinct phenotypic variants of the disease and progressing to differential diagnoses of athlete’s heart, hypertension, and infiltrative cardiomyopathies. HCM in children has its own section in this review, with possible risk factors that are distinct from those in adults; delayed enhancement in children may play a role in risk stratification in HCM. Finally, a number of teaching points for general cardiologists who recommend CMR for patients with HCM will be presented.

## 1. Introduction

Hypertrophic cardiomyopathy (HCM) is a primary diffuse or segmental left ventricular hypertrophy in the absence of secondary causes capable of producing hypertrophy. It is the most common congenital cardiac disease, having a prevalence of 1 in 500 people and representing an important cause of sudden death in adolescents [1]. HCM is an autosomal dominant disorder caused by mutations in the 11 sarcomeric genes, encoding heart sarcomere components [2,3]. The beta-myosin heavy chain gene, myosin-binding protein C gene, and troponin T gene are the most frequently mutated genes [4,5]. Genetic testing currently detects a known pathogenic or probable pathogenic mutation in 30–40% of patients with phenotypic HCM [6]. HCM is defined by a maximal wall thickness greater than 15 mm in general population (or greater than 13 mm in patients with a family history of HCM) and by a septal to left posterior wall thickness ratio greater than 1.3 in normotensive patients and greater than 1.5 in hypertensive patients [7,8]. In children, HCM is defined as an increase in LV wall thickness of more than 2 standard deviations from the mean value for age and BMI [9]. In most cases, LV hypertrophy is asymmetric, affecting mostly the interventricular septum, while hypertrophy can occur in a variety of places, including the apex, anterolateral wall, and free wall [10]. Left ventricular outflow tract (LVOT) obstruction, mitral valve apparatus abnormalities such as systolic anterior motion (SAM) or mitral regurgitation (MR), myocardial ischemia, myocardial fibrosis, and disarray are the key characteristics of HCM that explain the pathophysiology of the condition [11] (Figure 1). HCM can manifest clinically in a variety of ways, ranging from asymptomatic incidental echo findings to atrial or ventricular arrhythmias, syncope, severe heart failure, or even sudden cardiac death. Transthoracic echocardiography has traditionally been used to diagnose HCM, identifying disease characteristics such as left ventricular hypertrophy, systolic anterior motion of the mitral valve, and LVOT obstruction [12]. Risk stratification in HCM is based on personal history of aborted SCD, ventricular fibrillation, or sustained ventricular tachycardia; family history of SCD; and syncope. However, in recent years, CMR with presence of LGE has emerged as a risk marker for adverse outcomes. The implications of CMR in the detection and prognosis of HCM will be the focus of this review.

## 2. An Anatomical and Histopathological Review of Hypertrophic Cardiomyopathy

The microscopic alterations of the left ventricle in HCM have been recognized since 1957, when Donand Teare investigated [13] the causes of sudden death in eight individuals aged 14 to 44 who had nonspecific symptoms such as dyspnea, fatigue, and mild pectoral angina. On histopathological examination of hypertrophied hearts, the author noticed myocardial disarray, which consisted of an abnormal arrangement of muscle fibers and connective fibrous tissue, clefts and fissures, forming small channels between the two ventricles, preventing the heart from contracting efficiently.

The most prominent diagnostic aspects during a histological examination are: myocyte fiber disarray, interstitial, perivascular, and plexiform fibrosis, as well as small vessel disease, with dysplasia of small intramural coronary arterioles which manifest not only in fibrotic regions but also in areas surrounded by normal myocytes. Myofiber disarray is defined by the torsion of myocyte bundles, and also individual myocytes, with contractile components inside the sarcomere arranged perpendicularly or obliquely. Cardiac cells have larger, pleomorphic, and hyperchromatic nuclei. As a result, heart tissue contractility decreases, and this is compensated by cellular divisions.

In several studies LV hypertrophy was predominantly seen at the level of the interventricular septum and the anterior ventricular wall, also extending to the mitral valves and causing fibrosis. Sutton et al. analyzed the histopathology of 40 necropsy hearts from patients with HCM (10), congestive cardiomyopathy (10), aortic valve stenosis (10) and normal hearts (10) and evaluated the degree of fibrosis and fiber disarray using a four-stage grading system. The findings revealed that fiber disarray was significantly increased in all areas, despite the fact that fibrosis was not severe and the fiber disarray did not correlate with the wall thickness. Patients with hypertrophic cardiomyopathy showed plexiform fibrosis, however, this was not specific for the diagnosis [14]. Later, Maron et al. [15]. observed that myocyte disarray occurred in 88% of HCM patients during autopsy, 70% exhibiting myocardial fibrosis and 56% displaying thickened intramural arterioles with luminal narrowing. Maximal ventricular septal thickness ranged from 13 to 33 mm.

The LVOT becomes obstructed in late systole due to asymmetric hypertrophy of the septum, which generates a Venturi effect, leading to the dragging of the anterior mitral into the LVOT, a condition known as systolic anterior motion (SAM) of the mitral valve (Figure 2). Mitral regurgitation can occur when the AML is dragged towards the LVOT, by causing a space to appear between the mitral valve’s two leaflets. The mitral valve leaflets are elongated in HCM, especially the AML, which can be >30 mm long and thus be implicated in LVOT obstruction due to a steeper angle established between the AML and the septum. In 10% of HCM cases, there is a mid-cavity obstruction, and 25% of these patients develop LV apical aneurysm.

In a study on 1532 patients conducted by McLeod et al. [16] to better understand the cardiac features of this congenital disease, histological features identified included hyperchromatic nuclei in larger myocytes, myofibrillar disarray, interstitial and endocardial fibrosis. The curvature of the septum was used to classify the types of HCM: sigmoid septal morphology, reverse curve septal morphology, apical variant HCM, and neutral septal shape, with a weak association between the type of HCM and the degree of histological abnormality. Endocardial hypertrophy or mural plaque formation in the left ventricular outflow tract, as well as thickness of the anterior mitral leaflet, were characterized as hallmarks of HCM hearts that contributed to sudden death by Kocovski et al. [17]. According to Etfthimiadis et al. [18] outflow tract obstruction can occur not only in the subaortic area, but also in the mid-ventricular region (8%), resulting in an hourglass-shaped ventricle in 42.2% of patients. Overall, 26.5% of patients with midventricular obstruction had an apical aneurysm as a result of the elevated intracavitary pressure, and all of the obstructive HCM hearts’ left atria were dilated.

Hypertrophy was always present in Lamke’s study [19] of 204 septal myectomy HCM patients, with 100 of the 104 hearts exhibiting a significant subaortic septal bulge and four having an angled ventricular septum. In 79% of patients, myocardial disarray was present, and was more common in those under 65 years old. Fibrosis was found in 93% of the examined hearts, with 46% having abnormally thickened arteries. Galati et al. [20] observed three types of fibrosis: replacement fibrosis (53.3%), perimyocyte fibrosis (13.3%), and mixed (33.3%). The most common pattern was localized on the midwall, sometimes seen extending to the subepicardium or subendocardium. The upper anterior septum (36%) was the thickest region detected on echocardiography in Shapiro’s study [21], followed by the lower anterior septum (20%), upper posterior septum (9%), and the upper or lower free wall (21%).

## 3. Cardiac MR for HCM

Although echocardiography is an important tool in the evaluation of patients with left ventricular hypertrophy, it has certain limitations, especially in patients with an inadequate echocardiographic window. CMR bears an advantage in its capability to produce a more detailed three-dimensional image of the heart with high spatial and temporal resolution of the left ventricle. Additionally, it can differentiate between forms of HCM (Figure 3). The three main phenotypes of HCM are: asymmetric, concentric, and apical hypertrophy, with asymmetric septal being the most frequent between them (Figure 4). CMR can distinguish between different types of HCM. Moreover, CMR permits the visualization of the LV apex even in individuals with severe obesity.

CMR remains the gold standard for measurement of left ventricular volumes as well as systolic and diastolic functions. Furthermore, CMR presents the advantage of being able to reliably characterize myocardial tissue (Figure 5 and Figure 6). A maximum LV wall thickness higher than or equal to 15 mm in the end-diastolic phase is the most common diagnostic criteria for HCM. The relationship between late gadolinium enhancement and fibrotic zones has been studied extensively and has been associated with poorer clinical outcomes. Recently, T1 mapping techniques have improved, allowing for more detailed characterization of the diffuse fibrosis. For late gadolinium enhancement, 0.2 mmol/kg gadolinium contrast is injected at a speed of 1.5 mL/s. Ten minutes after infusion, acquisition begins, scanning the whole left ventricle from base to apex. Some categories of patients with HCM have a contraindication for CMR (Table 1). Myocardial fibrosis in HCM can be quantified as a percentage of the LV myocardium by using contrast-enhanced cardiac magnetic resonance with late gadolinium enhancement. Individuals with LGE >15% are at greater risk for sudden death [12,22,23].

Until recently, there were no methods to determine myocardial disarray in vivo, but this can now be assessed using diffusion tensor cardiac magnetic resonance (DT-CMR) imaging. The preferential diffusion of water along cardiac muscle fibers in HCM can be detected and compared to control cases, with the diffusion being quantified as fractional anisotropy [11]. HCM shows lower diastolic fractional anisotropy than control patients according to Ariga R et al. [24].

### 3.1. CMR Evaluation of Apical Hypertrophic Cardiomyopathy

Apical hypertrophic cardiomyopathy was initially described in 1976 in Japan, and it is thought to be a more benign form that affects 25% of HCM patients [28,29]. Apical HCM is more prevalent in men than women and the mean age at presentation is 40 years old [30]. CMR may be more effective than echocardiography in detecting early apical HCM. In 40% of patients, echocardiography missed apical hypertrophy, which was later discovered by CMR [31]. In HCM, late gadolinium enhancement is common; its presence and degree may be related to the severity of hypertrophy, as well as a higher risk of ventricular arrhythmia and sudden cardiac death [32]. The distribution of LGE in apical HCM is apical and subendocardial (Figure 7), a pattern that is unusual in other HCM variants [33]. In the absence of coexisting ischemic heart disease, LGE can be seen in up to 45.8% of patients with apical HCM [34].

### 3.2. CMR Evaluation of Focal Hypertrophic Cardiomyopathy

Localized hypertrophy affecting one or two LV segments is found in 20% of HCM patients, with the anterolateral LV wall, apex, or posterior septum being the most common locations. When HCM is suspected based on clinical symptoms, non-diagnostic electrocardiographic abnormalities, or family history, CMR should be used to confirm the diagnosis [36,37] when echocardiogram cannot. By identifying a focal pattern of hypertrophy, CMR can distinguish between athlete’s heart and HCM. Not only does deconditioning support a HCM diagnosis but so does the distribution of focal areas inside the left ventricle demonstrated by CMR [38]. Furthermore, late gadolinium enhancement has the potential to differentiate HCM from athlete’s heart due to its capacity to detect focal regions of fibrosis and increased extracellular volume [22].

Focal HCM should be differentiated from cardiac masses of the left ventricular wall. Since myocardial contractility is still present in HCM, and given that neoplastic tumors should not be contractile, CMR may distinguish HCM from other cardiac masses. Additionally, the signal intensity of tumor-related myocardial masses is usually different from that of normal myocardium. Specifically, CMR features of tumor-like HCM are more similar to those of neighboring normal myocardium [39,40].

### 3.3. CMR Evaluation of Midventricular Obstruction

Only 1% of HCM patients have associated midventricular obstruction [41]. This type of HCM is linked to mutations in the essential or regulatory light chains of the myosin heavy chain and is characterized by asymmetric left ventricular hypertrophy with high interventricular pressure gradients. Midventricular HCM is sometimes associated with an apical aneurysm caused by elevated systolic pressures in the apex due to midventricular obstruction. This variant of HCM is important because of its association with myocardial necrosis, cerebral and peripheral embolism, ventricular tachycardia, ventricular fibrillation and sudden cardiac death [42].

### 3.4. CMR Evaluation of Apical Aneurysm

Apical aneurysms occur in 2% of patients with HCM and 13% to 15% of patients with apical form of HCM [35].

The presence of an apical aneurysm in HCM is associated with a high rate of cardiovascular morbidity and mortality [43]. The exact mechanism of aneurysm formation remains unknown. Some of the possible mechanisms are: systolic mid-ventricular obstruction with increased velocities towards the apex, increased pressure at the LV apex with decreased coronary perfusion; increased oxygen demand due to increased wall stress and a decrease in oxygen supply due to hypertrophy and reduced myocardial capillary density; and microvascular dysfunction and a decrease in coronary reserve due to LV hypertrophy. All of these factors can result in silent myocardial infarction with subsequent apical aneurysm [44,45,46,47,48].

Compared to echocardiography, CMR provides a more reliable imaging method for detecting LV apical hypertrophy and aneurysm as it offers morphological and functional information as well as histological characteristics such as LGE, which corresponds to myocardial fibrosis [49].

Based on Rowin’s work [50] aneurysm size is defined as the maximum transverse dimension assessed in the four-chamber view at the end of systole, and aneurysms are further classified into small (20 mm), medium (20–40 mm), and large (>40 mm). Larger aneurysms bear a higher risk of arrhythmic events as well as higher risk for heart failure [51]. Previous research has shown that patients with apical aneurysms also have a higher risk of thrombus formation and embolic events, especially those with medium or large aneurysms. Therapeutic anticoagulation is indicated in patients with apical aneurysms for primary prevention of thromboembolism [52].

Extensive LGE has been shown in recent studies to be a risk predictor for HCM patients who are at increased risk for SCD and may be candidates for primary prevention of SCD with an ICD implant [53].

### 3.5. CMR Evaluation of LVOT Obstruction

Obstruction of the LVOT due to systolic anterior movement of the mitral valve is one of the causes of exercise intolerance and dyspnea in HCM. Since invasive methods such as surgical myectomy, alcohol septal ablation, or catheter septal ablation should be considered if pharmacological treatment fails, the presence and degree of LVOT obstruction influences the therapy. CMR sequences can identify anomalies that contribute to the obstruction, such as an aberrant anterior papillary muscle insertion or an elongated anterior mitral valve leaflet.

### 3.6. CMR Evaluation of Mitral Valve and Papillary Muscles

Owing to its good spatial and temporal resolution, wide field-of-view, and multi-planar imaging capabilities, CMR is well suited to the examination of the mitral valve and papillary muscles [54] (Figure 8). CMR has a high intrinsic soft tissue contrast that can be increased using gadolinium-based contrast agents, allowing for better tissue characterization. Moreover, CMR allows for dynamic evaluation of the mitral valve and papillary muscle contraction as well as measurement of fibrous tissue and muscle thickness and mass [55].

Previous research has found enlarged anterior and posterior mitral valve leaflets in patients with HCM relative to age-sex-and body size matched control subjects. Furthermore, 30% of HCM patients had elongation of the anterior and posterior mitral valve leaflets of more than 2SD [56,57]. The same authors identified a subpopulation of HCM patients that had a distinct phenotype, with moderate hypertrophy and disproportionately enlarged mitral valves with significant leaflet elongation. Elongated mitral valve leaflets are frequently associated with LVOT obstruction [58]. For these reasons, in extremely symptomatic obstructive HCM patients, the combination technique of septal myectomy and anterior mitral leaflet repair is recommended [59].

Kaple et al. [60] characterized the morphology and histology of the mitral valve, finding that 80 of the 115 patients who underwent mitral valve surgery had restrictive leaflet abnormalities and 64 had elongated leaflets. Furthermore, 22 and 23 individuals had subvalvular chordal and papillary muscle abnormalities, respectively, with the muscles’ inappropriate attachments to the mitral valve causing systolic anterior movement, fibrosis, and left ventricular outflow obstruction.

Papillary muscles in HCM patients are usually hypertrophied, having a mass nearly twice that of papillary muscles in healthy controls. LGE caused by interstitial fibrosis can also be found in hypertrophied papillary muscles, usually in a patchy distribution in the mid-myocardial region and frequently near RV insertion locations. LGE of the papillary muscles is associated with more severe symptoms, cardiac dysfunction, and ventricular dysrhythmias [61,62]. Patchy regions of LGE in the papillary muscles can be detected on CMR in 6% of patients with HCM due to interstitial fibrosis. These patients have a higher papillary muscle mass than patients without LGE [55]. Papillary muscle hypertrophy has been associated with LV wall thickness and myocardial mass and the papillary muscle mass index has been associated with the degree of outflow gradient. These findings show that papillary muscle hypertrophy may be secondary to LV pressure overload produced by LV obstruction rather than being caused by a primary genetic defect. In patients with HCM, the distance between the papillary muscle and the septum is also a topic of attention, as this distance was found to be smaller in individuals with obstruction. Furthermore, individuals with HCM contain a larger number of papillary muscles than controls, with half of the patients bearing more than 3 papillary muscles [55,63].

### 3.7. CMR Evaluation of Left Atrial Dimensions and Function

CMR is a valuable tool for characterizing the left atrium since it can show not only the atrial shape but also the tissue composition, especially when the late gadolinium enhancement technique is applied. The volume of the left atrium can be measured in short-axis view using Simpson’s disc method or in the horizontal and vertical axes using the biplane area-length approach. In a four-chamber view, the area of the left atrium (LA) can be measured by tracing the outer line. CMR can distinguish between thrombi and tumors based on tissue composition, as the latter exhibits a hyperintensity on T2-weighted images, rendering it a useful tool for detecting hypertrophic cardiomyopathy complications at the level of the left atrium.

CMR, according to Kuchynka et al. [65] is one of the best instruments for evaluating LA volume, with the smallest volume being the most relevant for predicting cardiac complications. The CMR reference values for LA volumes are 97 ± 27 mL for men and 89 ± 21 mL for women. In addition, a normal LA area should not exceed 23 cm^2^. The enlargement of the LA, as well as its dysfunction, occurs prior to the development of heart failure, according to a study on atherosclerosis, however, further research is needed to characterize it for HCM.

Despite the fact that 2D echocardiography is the gold standard for assessing LA function, CMR can also be used to assess LA volumes during the cardiac cycle and calculate the emptying fraction. LA contribution to LV filling decreases with age while passive LA volume increases. The importance of CMR was proven in a study comparing 10 patients with HCM to 10 healthy individuals by examining the LA conduit, reservoir, and contractile functions.

For patients who develop atrial fibrillation, CMR can also be employed as a guiding tool during radiofrequency ablation. In patients with HCM, atrial fibrillation is the most common arrhythmia. The LA must be thoroughly characterized by CMR in order to detect its appearance and administer prophylactic or therapeutic medication. Kim KJ et al. conducted a blinded study to compare the morphological and functional differences in the LA between HCM types. When comparing CMR characteristics, the HCM group had lower end diastolic volume and end systolic volume, as well as higher ejection fraction and LV mass index than the control group. Only the LV maximal wall thickness, LV mass index, and the existence and degree of LGE were relevant across the different types of HCM. Accordingly, the authors showed that HCM patients have a larger LA, poorer reservoir function, and lower LA global strain than healthy individuals. LGE-MRI analysis revealed no statistical differences in LA remodeling and dysfunction between HCM phenotypes and LV myocardial fibrosis.

## 4. CMR Differential Diagnosis of Thickened Myocardium

### 4.1. Differential Diagnosis with Athlete’s Heart

Athlete’s heart refers to morphologic modifications of the heart caused by intense physical activity: increased LV mass, increased LV diastolic diameter, and LV wall thickness. The absence of areas of delayed enhancement of the LV myocardium is a key aspect of the cardiac remodeling seen in athletes.

Athletes may acquire ventricular hypertrophy because of regular training, and it can be difficult to distinguish from HCM [66]. Histologically, the athlete’s cardiac hypertrophy is the result of cellular hypertrophy, whereas HCM is characterized by cellular disarray and extracellular expansion. When ECG and echocardiogram are inconclusive, CMR is an effective technique for distinguishing HCM from athlete’s heart, especially in patients with poor acoustic windows and hypertrophy affecting the apex, the basal antero-lateral wall, the inferior septum, or the inferior RV.

Luijkx et al. [67] conducted a study with the purpose of defining the characteristics of athlete’s heart and HCM, even when borderline hypertrophy (septal wall thickness values between 12 and 16 mm in males and 11 and 16 mm in women) was present. The researchers performed a retrospective study in which they examined various types of athletes and patients with HCM, considering LV diameters and ECV. The authors established a formula that can distinguish 99.5 percent of HCM patients from athletes’ heart (AUC = 0.995). As a result, as compared to both athletic and non-athletic healthy controls, the ratio of LV EDV (End Diastolic Volume) to LV EDM (End Diastolic Mass) was lower in patients with HCM. This model was still effective for borderline individuals (AUC = 0.992). This algorithm can be used to predict the progression of ventricular hypertrophy in various groups. Disproportional hypertrophy is a feature of HCM, with EDM growing as EDV decreases. Physiological hypertrophy, such as that seen in athletes, is proportionate, with both the EDM and EDV growing at the same time, resulting in increased heart strength and the ability to pump efficiently even under physical exertion.

### 4.2. Differential Diagnosis with Hypertensive Heart Disease

Hypertensive heart disease (HHD) is a disease that develops because of high blood pressure persisting for a long period of time. In contrast to HCM, which is characterized by myocyte disarray and focal interstitial fibrosis, hypertensive heart disease is defined by left ventricular hypertrophy, proportionate hyperplasia, and diffuse interstitial fibrosis. Since both diseases have concentric asymmetric left ventricle hypertrophy (LVWT 15 mm for HCM and 12 mm for HHD), LGE, reduced LV strain, and diastolic dysfunction, the type and quantity of fibrosis is the only way to differentiate them.

The effectiveness of T1 mapping for discriminating between HHD and HCM was investigated in a double-blinded retrospective study by Neisus et al. [68]. They examined 232 individuals, 108 of whom had been diagnosed with HCM and 53 with HHD. Patients with HCM and HHD were matched by gender, presence of LVH, and maximal LVWT, whereas those with similar age, LV mass index, and global and septal T1 were grouped together. In HCM patients, global native T1 mapping was significantly higher than global native T1 (108,631 vs. 110,439 ms). Global native T1 and LV mass index or maximal LVWT were mildly correlated only in HCM patients.

Hinojar et al. [69] conducted another study to assess the role of CMR in distinguishing HCM from HHD. The authors divided the studied population into 4 groups, diagnosed HCM = 95, diagnosed HHD = 69, carriers of the HCM gene mutation = 23 and healthy non-athletic controls = 23. They used a CMR technique which included LGE, T1 mapping, and additional mass and tissue characterization. Both LVH groups had diastolic dysfunction, with HCM patients having a larger LV mass and a higher LVWT. LGE was found not only in HCM participants, but also in HHD patients on rare occasions, with the latter showing predominantly an ischemic pattern of LGE.

Takeda et al. [70] conducted a retrospective study to investigate the use of CMR to differentiate between forms of cardiomyopathies. Twenty-six participants underwent CMR, including Cine-MRI and LGE, as part of the study. Six were identified with cardiac amyloidosis, nine with end-stage HCM, and eleven with HHD. Patients with HHD had a higher LVEF than those with HCM, and pericardial effusions were more frequently seen in amyloidosis. In end-stage HCM, the number of LGE segments was higher than in HHD. LGE patterns in HCM were mostly observed in the anteroseptal or inferoseptal segments and they were mostly found to have patchy midwall or patchy epicardial patterns. On the other hand, midwall linear, midwall or epicardial LGE patterns were the most common in HHD, and were generally located in the septal to inferior areas of the midventricular level. The most useful type of CMR for establishing the diagnosis was Cine-MRI. LVEF was increased in HCM, but HHD was defined by progressive systolic dysfunction and a dilated LV cavity.

### 4.3. Differential Diagnosis with Infiltrative Cardiomyopathies

Restrictive cardiomyopathies such as amyloidosis, Fabry disease, glycogen storage disorders, sarcoidosis, iron overload, and myocardial fibrosis are difficult to distinguish from HCM since they have similar clinical characteristics and are defined by LVH [71] (Figure 9).

Amyloid infiltration of the ventricular wall causes hypertrophy. Amyloidosis can be differentiated from other restrictive cardiomyopathies such as HCM, HHD, and diastolic heart failure with LV hypertrophy using CMR. In contrast to amyloidosis, which has a normal or decreased ventricular contraction, HCM has a hyperdynamic contraction. In addition, amyloid fibrils tend to accumulate in the subendocardium, leading to interstitial expansion and the distinctive “zebra pattern” of LGE. Nam et al. [72]. conducted a retrospective blind study to assess CMR’s role in differentiating amyloidosis from HCM. A total of 46 amyloidosis patients, 30 HCM patients, and 10 asymptomatic subjects were included in the study. They were all exposed to CMR with and without gadolinium contrast injection. The native T1 blood pool of the myocardium in amyloidosis was the highest of the two groups, while the native T1 blood pool of the LV cavity was the lowest. In amyloidosis, ECV was also higher.

Although cardiac amyloidosis exhibits a distinct LGE pattern, this approach is often unavailable due to the presence of renal impairment in most amyloidosis patients. Therefore, Jung et al. examined various forms of tissue tracking analyses in order to determine the best native CMR approach for differentiating between amyloidosis and HCM. The retrospective study comprised of 54 patients diagnosed with amyloidosis, 40 HCM patients, and 30 healthy controls. The authors suggested specific CMR techniques for amyloidosis detection, with amyloidosis patients displaying relative apical sparing and much better ratios of wall thickness to LVEF. Moreover, T1 mapping was found to be a very effective first step in distinguishing these two diseases (AUC = 0.880).

In sarcoidosis CMR, provides a thorough assessment of cardiac structure and function, as well as an amplification of the mediastinal lymph nodes, allowing for a precise diagnosis. LGE is also evident, in cardiac sarcoidosis.

CMR may occasionally reveal another source of LV hypertrophy in HCM patients: glycogen storage disorders, the most prevalent of which is Fabry disease. This condition is characterized by a gradual intracellular accumulation of neutral glycosphingolipids, which can occasionally cause LVH, particularly when alpha-galactosidase A, the implied lysosomal enzyme, has only partial activity. Since the pattern of LGE in this condition, which represents sphingolipid accumulation and interstitial expansion due to myocardial fibrosis, differs from that in HCM or HHD, CMR can be employed to diagnose it.

## 5. CMR for Risk Stratification in Adults with HCM

Late gadolinium enhancement is currently not a risk classification criterion in European or American guidelines. In European Guidelines, HCM Risk-SCD Calculator accounts for factors such as age, presence of LV hypertrophy, left atrial size, LV outflow tract gradient, family history of SCD, nonsustained ventricular tachycardia, and idiopathic syncope [74]. In American guidelines, personal history of aborted SCD, ventricular fibrillation, or sustained ventricular tachycardia; family history of SCD; and syncope are all considered relevant risk factors [75]. Nonsustained ventricular tachycardia on monitoring, severe increase in left ventricular (LV) wall thickness (>30 mm), and hypotensive response to exercise are all markers with less convincing evidence.

However, in recent years, LGE has emerged as a risk marker for unfavorable HCM outcomes. LGE may be present in around half of all HCM patients. Nevertheless, the presence of LGE cannot be considered as a sole indicator of an ICD. The extent of LGE may be a more powerful indicator than its mere presence. A study [76] on 1293 patients with HCM followed for 3.3 years found that not just the presence of LGE but the extent of >15% of LV mass was associated with a two-fold increase in the probability of SCD.

## 6. Special Considerations in Children

HCM is responsible for 42% of cardiomyopathy in children, and annual HCM-related mortality is close to 1% [77,78]. In children and young individuals, HCM is a common cause of arrhythmias and sudden cardiac death [79]. Pediatric HCM encompasses a more diverse group of abnormalities when compared with adult HCM. HCMs with sarcomeric gene abnormalities are called primary HCMs, and non-sarcomeric gene abnormalities are called secondary HCMs.

Most of them, 40–60%, are due to mutations in the sarcomeric genes. Other causes of HCM include inherited metabolic errors such as glycogen storage disorders (Pompe disease, Danon disease, Cori-Forbes disease, PRKAG2 syndrome), lysosomal storage diseases (mucopolysaccharidoses), fatty acid oxidation disorders, endocrine disorders, neuromuscular diseases, mitochondrial diseases (Barth syndrome, Friedreich ataxia), and malformation syndromes (Noonan syndrome, Costello syndrome, cardiofaciocutaneous syndrome, neurofibromatosis Type 1, Legius Syndrome) which account for up to 35% HCM cases in children [80,81].

While echocardiography remains the gold standard for diagnosing HCM in children, cardiovascular magnetic resonance imaging can be useful in pediatric patients with diagnostic uncertainty, such as those with a suspected metabolic or lysosomal storage disorder or malformation syndrome, in case of poor echocardiographic imaging windows, or as an adjunct in assessing the risk of sudden cardiac death by assessing late gadolinium enhancement in children with confirmed HCM [82].

## 7. Risk Stratification in Children with HCM

In contrast to adults, in children there is a non-linear relationship between the grade of LVH and the risk of SCD. As a result, there is no absolute threshold LV wall thickness at which an ICD should be recommended for primary prevention. Compared with adults, in children, the LV outflow tract gradient and a familial history of SCD are not associated with an increased risk of SCD. LGE occurs in children with HCM, and the extent of LV LGE is similar to that in adults. Moreover, LGE is associated with adverse events in children with HCM: decreased ejection fraction, degree of heart failure, ventricular arrhythmias and sudden cardiac death. Therefore, an extensive late gadolinium enhancement in CMR imaging could be used to improve risk stratification in children with an unclear risk level [83]. It is worth noting that LV apical aneurysms develop over time in children with midventricular form of HCM, therefore, aneurysms with LGE are more common in older children [43].

## 8. Conclusions

CMR has evolved as a remarkable imaging method suited for the characterization of varied phenotypes in HCM. CMR can provide diagnostic and prognostic information that conventional echocardiogram cannot. Late gadolinium enhancement can be used to distinguish between HCM and other disorders that mimic it. CMR bears an impact on the clinical management from screening to diagnosis and best therapy option.

## 9. Teaching Points

CMR allows accurate thickness measurements of the LV walls.Localized or diffuse myocardial fibrosis inside the left ventricle can be identified with CMR.CMR can be used to describe the various phenotypic manifestations of sarcomeric gene mutations.CMR can identify other issues associated with HCM, such as apical aneurysm, mitral valve and papillary muscle anomalies, LV outflow tract obstruction, midventricular obstruction, and left atrial enlargement.CMR with late gadolinium enhancement can be used to differentiate HCM from hypertensive heart disease, athlete’s heart and infiltrative cardiomyopathies.In children, late gadolinium enhancement might be used to improve risk stratification, as LV outflow tract gradient and family history of SCD are not associated with an elevated risk of SCD, and there is a non-linear relationship between LVH and the risk of SCD.

## Figures and Tables

**Figure 1 diagnostics-12-00489-f001:**
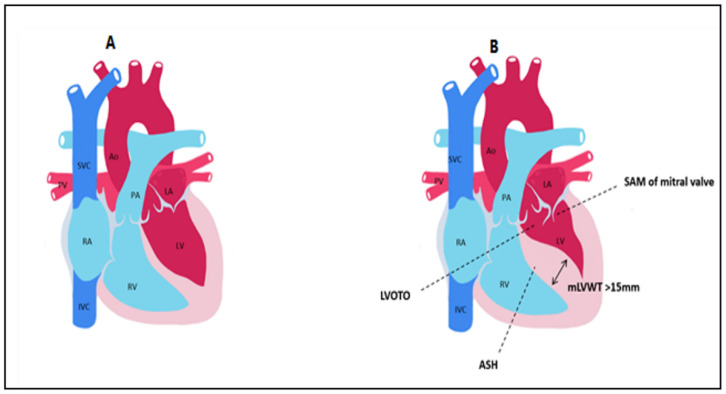
Main anatomical aspects found in hypertrophic cardiomyopathy compared to control. (**A**) normal heart. (**B**) HCM LVOTO = Left ventricular outflow tract obstruction; SAM = systolic anterior motion; ASH = asymmetric septal hypertrophy; mLVWT = maximal left ventricular wall thickness; Ao = aorta; PA = pulmonary artery; LA = left atrium; LV = left ventricle; PV = pulmonary veins; RA = right atrium; RV = right ventricle; SVC = superior vena cava; IVC = inferior vena cava.

**Figure 2 diagnostics-12-00489-f002:**
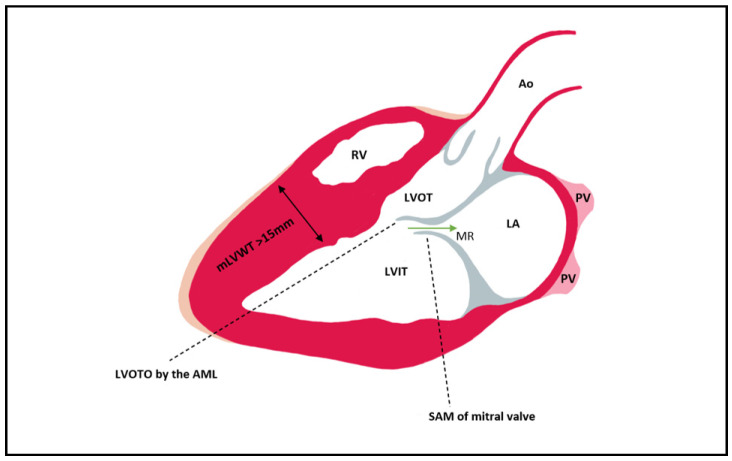
Left ventricular outflow tract obstruction by the anterior mitral leaflet which is pulled during late systole into the LVOT as a consequence of the Venturi effect. In most cases of HCM, the hypertrophy is mainly seen at the level of the septum. Mitral regurgitation can be present due to the SAM of AML that leads to a gap between the AML and PML (its presence can be established via Doppler). Ao = aorta; PV = pulmonary vein; LA = left atrium; LVIT = left ventricle inflow tract; LVOT = left ventricle outflow tract; RV = right ventricle; MR = mitral regurgitation; mLVWT = maximal Left ventricular wall thickness; LVOTO = Left ventricular outflow tract obstruction; SAM = Systolic anterior motion; AML = anterior mitral leaflet; PML = posterior mitral leaflet.

**Figure 3 diagnostics-12-00489-f003:**
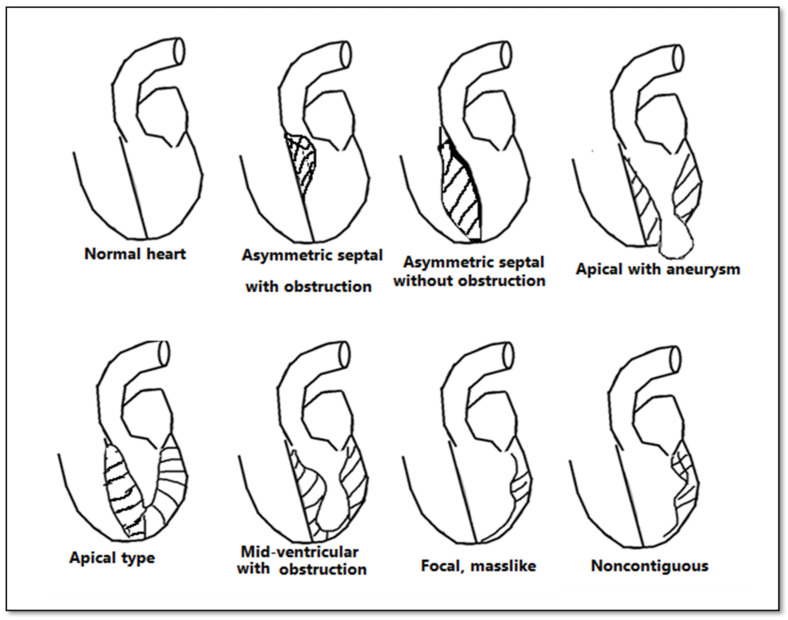
Anatomical types of HCM in function of the distribution of hypertrophy.

**Figure 4 diagnostics-12-00489-f004:**
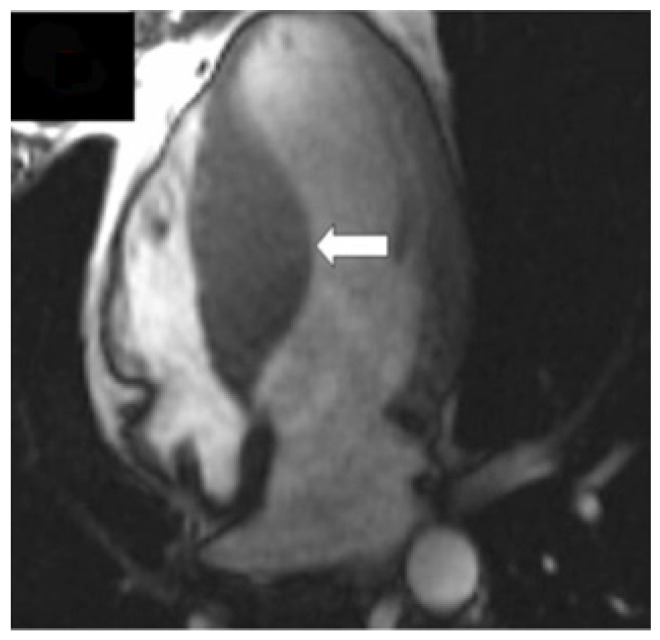
Cardiac magnetic resonance findings in a patient with asymmetric basal septal hypertrophy (white arrow). Reproduced with permission from [25]. Copyright 2015 Houston BA et al. (under the Creative Commons Attribution 4.0 License, https://creativecommons.org/licenses/by/4.0/, accessed on 6 February 2022).

**Figure 5 diagnostics-12-00489-f005:**
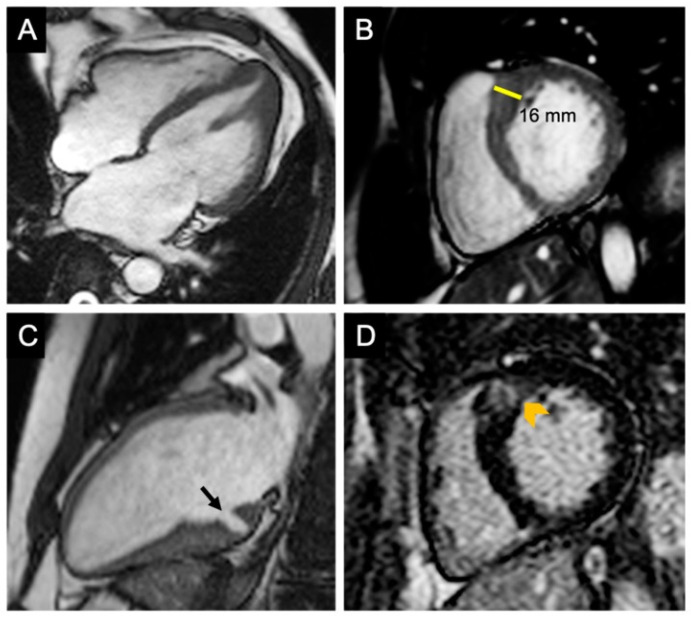
CMR imaging in the evaluation of HCM. (**A**) Four-chamber view shows hypertrophy of the apex and papillary muscle found in a patient with apical variant of HCM. (**B**) Short-axis view with hypertrophy of the junction between the basal anterior interventricular septum and LV anterior free wall; considered most usual location of hypertrophy in HCM. (**C**) Apical two-chamber view showing a myocardial crypt (marked with black arrow) at the base of the LV inferior wall. (**D**) LGE in short-axis view of the same patient from Figure 5B, demonstrating patchy replacement fibrosis, (marked with arrow). Reproduced and adapted with permission from [26]. Copyright 2019 Popa Fotea NM et al. (under the Creative Commons Attribution 4.0 License, https://creativecommons.org/licenses/by/4.0/, accessed on 6 February 2022).

**Figure 6 diagnostics-12-00489-f006:**
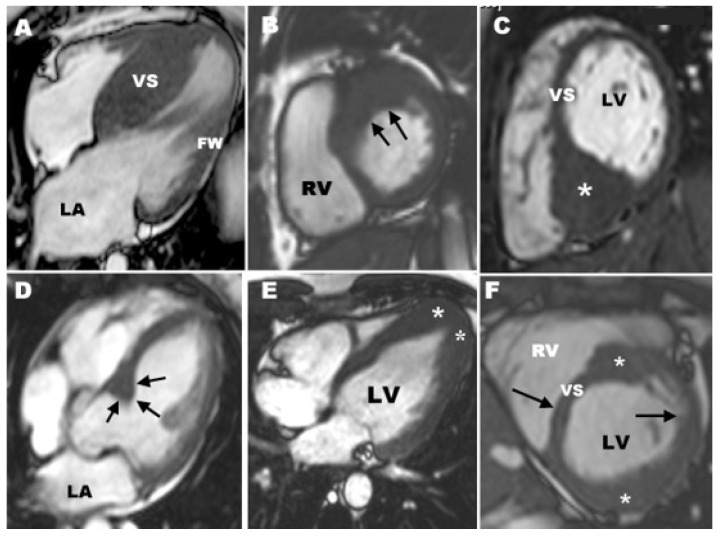
CMR during end-diastole showing different types of HCM. (**A**) Hypertrophy if the inter ventricular septum (VS), sparing the free wall of the LV (FW); (**B**) Hypertrophy of the LV free wall in the basal portion and a part of the contiguous anterior interventricular septum, which is the most common area involved in HCM; (**C**) Important hypertrophy >33 mm of the posterior interventricular septum (marked with an asterisk); (**D**) Focal HCM localized at the bass of the anterior interventricular septum (marked with arrows); (**E**) localized HCM at the LV apex (marked with asterisks); (**F**) segmental HCM with hypertrophy of the basal anterior interventricular septum and anterolateral LV wall (marked with asterisks), separated by normal LV (arrows). Reproduced and adapted with permission from [27]. Copyright 2012 Elsevier, Maron MS et al.

**Figure 7 diagnostics-12-00489-f007:**
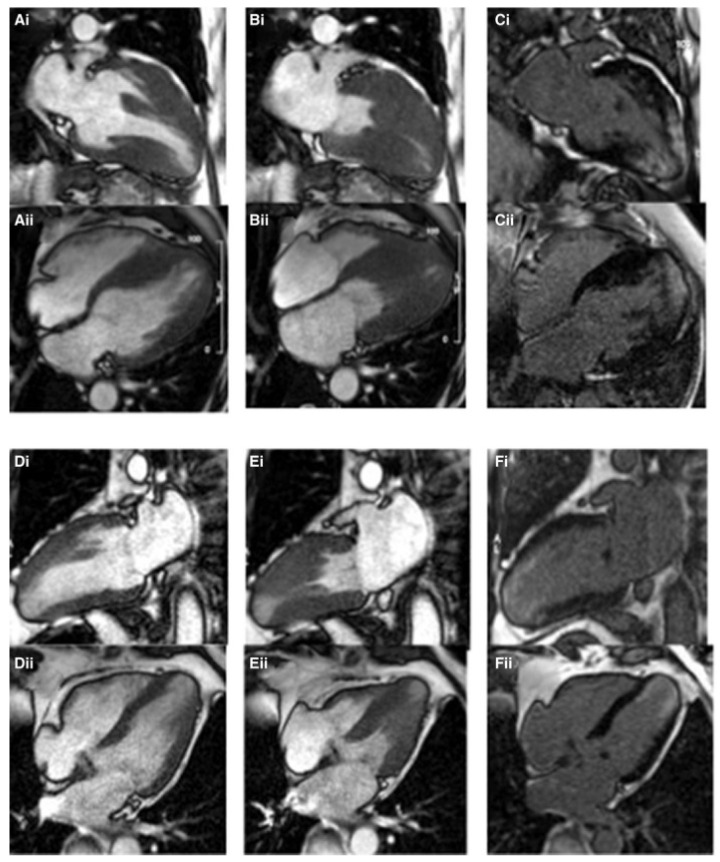
CMR comparison of apical aneurysm formation in mixed Apical HCM (**A**–**C**) versus pure Apical HCM (**D**–**F**). Long axis images of a patient with mixed Apical HCM during diastole in two chambers (**Ai**) and four chambers (**Aii**) views, which in systole show midventricular obstruction but not total cavity obliteration due to apical chamber persistence (**Bi**,**Bii**). LGE is found in the apical aneurysm (**Ci**,**Cii**). A thinned aneurysmal apex is visible in diastole on 2 (**Di**) and 4 chamber views in a separate patient with pure apical HCM (**Dii**). The apical aneurysm (**Ei**,**Eii**) becomes visible in systole and contains LGE (**Fi**,**Fii**). Reproduced and adapted with permission from [35]. Copyright 2020 Hughes RK et al. (under the Creative Commons Attribution 4.0 License, https://creativecommons.org/licenses/by/4.0/, accessed on 6 February 2022).

**Figure 8 diagnostics-12-00489-f008:**
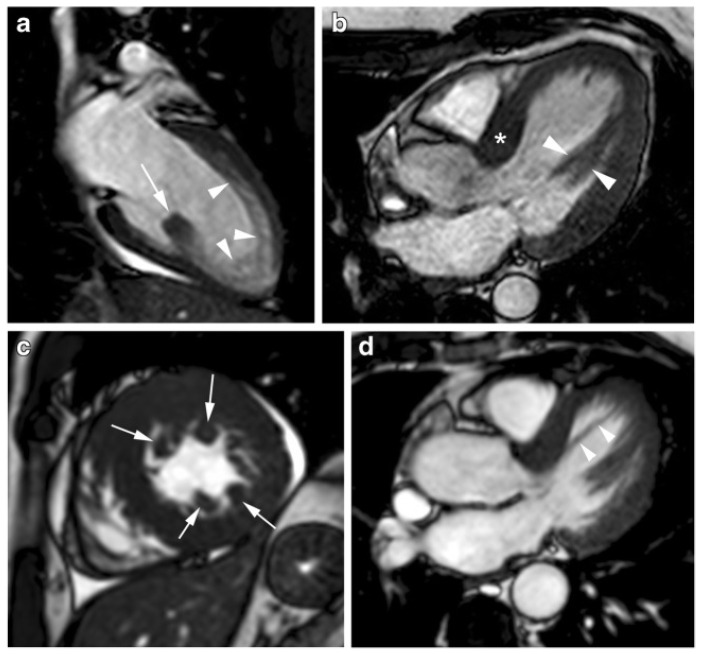
Abnormalities of mitral valve and papillary muscles in HCM. (**a**) Isolated posterior papillary muscle hypertrophy (marked with an arrow) and trabeculations along the anterior wall and apical wall (marked with arrowheads); (**b**) Bifid anterolateral papillary muscle (marked with arrowheads) and thickening of the basal septum (marked with an asterisk); (**c**) Four accessory papillary muscles, (marked with arrows), diffuse hypertrophy of the left ventricle and pericardial effusion; (**d**) Accessory apical muscle bundle (marked with arrowheads) traverse the cavity of the left ventricle from the septal basal wall to the distal part of the LV. Reproduced and adapted with permission from [64]. Copyright 2015 Soler R et al. (under the Creative Commons Attribution 4.0 License, https://creativecommons.org/licenses/by/4.0/, accessed on 6 February 2022).

**Figure 9 diagnostics-12-00489-f009:**
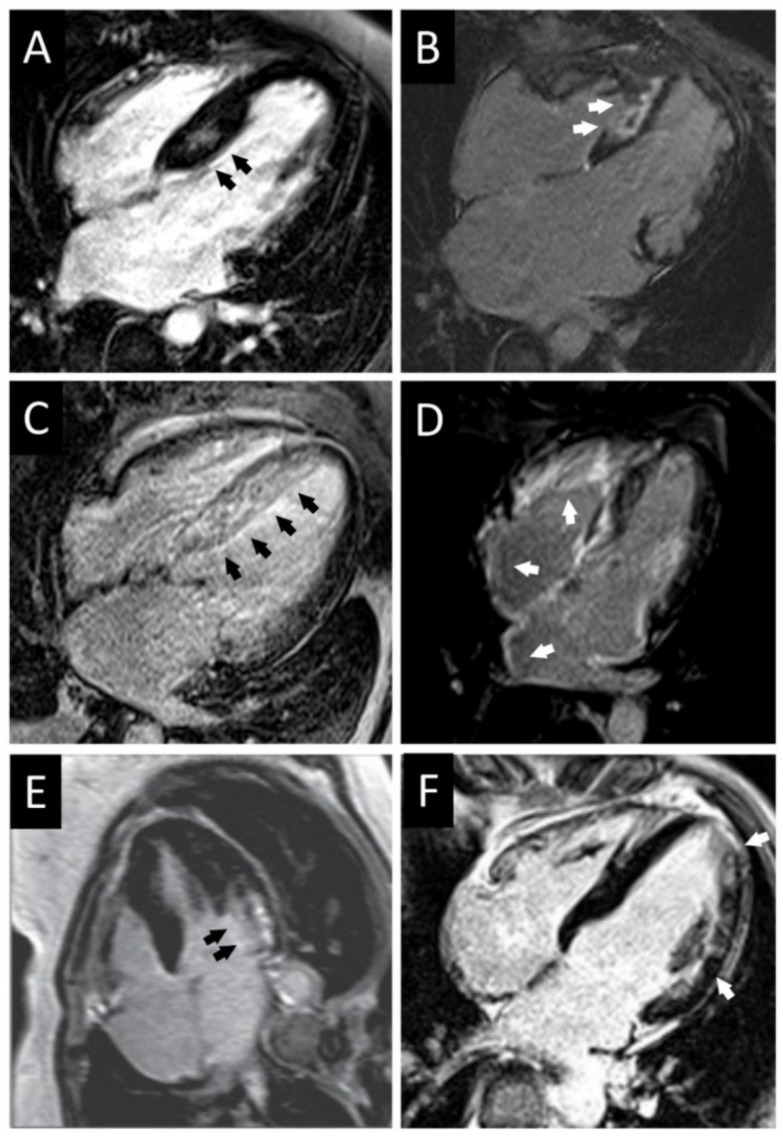
Findings on cardiac magnetic resonance in patients with: (**A**) Obstructive HCM with intramural septal LGE (arrows: intramural septal LGE). (**B**) HCM with restricted physiology, significant atrial enlargement, and substantial septal fibrosis (arrows: septal fibrosis). (**C**) Transmural septal LGE with AL amyloidosis (arrow shows septal LGE). (**D**) Amyloidosis of wtATTR with LGE, especially in the right ventricle and atria (arrows: LGE in both the right ventricle and left and right atrium). (**E**) Fabry disease; a case of hypertrophic phenotype with subendocardial LGE in the basal lateral region of the left ventricle (arrows: LGE in the left ventricle’s basal lateral region). (**F**) Fabry disease with moderate hypertrophy and intramural LGE in the left ventricle’s mid-lateral segment and apex (arrows: LGE in the left ventricle’s mid-lateral segment and apex). Reproduced and adapted with permission from [73]. Copyright 2021 Vio R et al. (under the Creative Commons Attribution 4.0 License, https://creativecommons.org/licenses/by/4.0/, accessed on 6 February 2022).

**Table 1 diagnostics-12-00489-t001:** Contraindications for CMR.

Implanted ICDs *
implanted pacemakers *
brain ferromagnetic clips
cochlear implants
metal foreign body (bullet fragments, metallic splinter in the eye)
claustrophobia

* CMR has a relative contraindication in patients with an implanted cardiac device. MR Safe or MR Conditional devices, on the other hand, can be scanned using the proper protocol.

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
