# Peer review of "Anatomical-MRI Correlations in Adults and Children with Hypertrophic Cardiomyopathy"

_diagnostics, 2022, doi:10.3390/diagnostics12020489_

Round 1

Reviewer 1 Report

Nice topic for the cardiac imaging.

I think it would be good to improve the syntax and form of the text. 

Author Response

We would like to thank all 3  reviewers for their thoughtful review of the manuscript. They raise important issues and their inputs are very helpful for improving the manuscript. We agree with all their comments and we have revised our manuscript accordingly.

We marked with red color the modifications we have made in the revised manuscript.

Please, find below the referees’ comments repeated and our responses inserted in after each comment.

REVIEWER 1.   COMMENTS TO AUTHOR:

Nice topic for the cardiac imaging.

I think it would be good to improve the syntax and form of the text.

Response: The form and syntax were improved by doctor Julien Redfern, a native English speaker from London-Great Britain.  

Reviewer 2 Report

The topic is not really new and already discussed enough in the literature,
however the manuscript is well done.
My advice is to try to expand the part relating to pediatric patients
to improve the work.

Author Response

We would like to thank all 3  reviewers for their thoughtful review of the manuscript. They raise important issues and their inputs are very helpful for improving the manuscript. We agree with all their comments and we have revised our manuscript accordingly.

We marked with red color the modifications we have made in the revised manuscript.

Please, find below the referees’ comments repeated and our responses inserted in after each comment.

REVIEWER 2.   COMMENTS TO AUTHOR:

The topic is not really new and already discussed enough in the literature,
however the manuscript is well done.
My advice is to try to expand the part relating to pediatric patients
to improve the work.

Response: We expanded the part related to pediatric patients. Please find the modifications highligted in red in the improved version of the manuscript

Reviewer 3 Report

The introduction could use some basic info such as in how many % of phenotype positive patients are genotype positive, as well as an introduction to risk stratification. The remaining of the manuscript often repeats small parts as ‘LGE is associated with ventricular arrhythmias’, but the authors should introduce these concepts in the beginning. This also provides clear evidence of the reason why CMR is that important in HCM.

A separate section of risk stratification in HCM with paragraphs for adults and children would be useful. This can help avoiding unnecessary repetitions such as: LGE is associated with SCD. For each of the separate entities described in section 3, LGE will be associated with SCD/arrhythmias. Therefore introducing this early on will improve the reading of other sections.

There are multiple issues with the references, both the formatting as absence of references. Please cite the study after ‘… et al.’, for example ‘Takeda et al.74 conducted’

The teaching points and conclusion should be separate.

Minor:

References: the comma in between references are not in superscript.

Occasionally there are double spaces throughout the abstract and manuscript.

The resolution of Figure 1 is too low for adequate interpretation.

Line 103: please provide units to ’13 to 33’.

Line 113: Reference to McLeod et al. is missing

Line 118: ‘weak’ instead of ‘week’

Line 121: Reference to Kocovski et al is missing.

Line 127: Reference to Lamke’s study is missing.

Section 3 requires editing:

  • Line 143 and line 147 contain repetitions
  • References are missing for the statements in line 148-158.
  • The contra-indications for CMR are lost in the text and could benefit from a table for example.
  • Line 163-164: This sentence is out of place and requires more information. It also copies the information in line 157-158. The why and how should be stressed into more detail.
  • Figure 3 is poor quality and this important figure should be edited significantly or be replaced with CMR examples of these such as these of Figure 5 and 6.
  • Line 354: 708%?
  • The LA section is quite extensive and contains some unnecessary information. Whereas it does not mention that 2D LA diameter on echo is used in SCD risk stratification, nor does it clearly state what CMR can add on top of that.

Section 4:

  • Line 383: reference is missing
  • Line 401: Luijkx et al. reference is missing
  • Line 418: ‘id’ is ‘is’
  • Amyloidosis does not require a capital A each time.

Section 5:

  • Excessive information in sentences with differential diagnoses. This could be improved by presenting this information in a table with incidences/prevalences.
  • Line 514 suddenly presents the information on risk stratification in adults… as mentioned above… a separate section in the beginning on risk stratification in HCM, both adults and children, can avoid unnecessary repetition.

Author Response

We would like to thank all 3  reviewers for their thoughtful review of the manuscript. They raise important issues and their inputs are very helpful for improving the manuscript. We agree with all their comments and we have revised our manuscript accordingly.

We marked with red color the modifications we have made in the revised manuscript.

Please, find below the referees’ comments repeated and our responses inserted in after each comment.

REVIEWER 3.   COMMENTS TO AUTHOR:

The introduction could use some basic info such as in how many % of phenotype positive patients are genotype positive, as well as an introduction to risk stratification. The remaining of the manuscript often repeats small parts as ‘LGE is associated with ventricular arrhythmias’, but the authors should introduce these concepts in the beginning. This also provides clear evidence of the reason why CMR is that important in HCM.

  • Response: Thank you. We made changes to the introduction section based on your feedback.

A separate section of risk stratification in HCM with paragraphs for adults and children would be useful. This can help avoiding unnecessary repetitions such as: LGE is associated with SCD. For each of the separate entities described in section 3, LGE will be associated with SCD/arrhythmias. Therefore introducing this early on will improve the reading of other sections.

  • Response: Thank you.. There are now 2 distinct sections on risk stratification for adults and children.

There are multiple issues with the references, both the formatting as absence of references. Please cite the study after ‘… et al.’, for example ‘Takeda et al.74 conducted’

  • Response: Thank you. The references have been updated.

The teaching points and conclusion should be separate.

  • Response: Thank you. They were separated.

Minor:

References: the comma in between references are not in superscript.

Occasionally there are double spaces throughout the abstract and manuscript.

The resolution of Figure 1 is too low for adequate interpretation.

Line 103: please provide units to ’13 to 33’.

Line 113: Reference to McLeod et al. is missing

Line 118: ‘weak’ instead of ‘week’

Line 121: Reference to Kocovski et al is missing.

Line 127: Reference to Lamke’s study is missing.

  • Response:  Thank you. The double spaces were corrected and references have been updated.

Section 3 requires editing:

  • Line 143 and line 147 contain repetitions
  • References are missing for the statements in line 148-158.
  • The contra-indications for CMR are lost in the text and could benefit from a table for example.
  • Line 163-164: This sentence is out of place and requires more information. It also copies the information in line 157-158. The why and how should be stressed into more detail.
  • Figure 3 is poor quality and this important figure should be edited significantly or be replaced with CMR examples of these such as these of Figure 5 and 6.
  • Line 354: 708%?
  • The LA section is quite extensive and contains some unnecessary information. Whereas it does not mention that 2D LA diameter on echo is used in SCD risk stratification, nor does it clearly state what CMR can add on top of that.

  • Response: We made the necessary changes to the text.

Section 4:

  • Line 383: reference is missing
  • Line 401: Luijkx et al. reference is missing
  • Line 418: ‘id’ is ‘is’
  • Amyloidosis does not require a capital A each time.
  • Response: Thank you. We made the necessary changes.

Section 5:

  • Excessive information in sentences with differential diagnoses. This could be improved by presenting this information in a table with incidences/prevalences.
  • Response: The information has been condensed.
  • Line 514 suddenly presents the information on risk stratification in adults… as mentioned above… a separate section in the beginning on risk stratification in HCM, both adults and children, can avoid unnecessary repetition.
  • Response: There is now a distinct section on risk stratification for adults and children.

Round 2

Reviewer 1 Report

As described above, I think the issue is very relevant.
However, I believe a moderate review of English language and form is essential. 

Author Response

We would like to thank the 2  reviewers for their thoughtful review of the manuscript. They raise important issues and their inputs are very helpful for improving the manuscript. We agree with all their comments and we have revised our manuscript accordingly.

We marked with red color the modifications we have made in the revised manuscript.

Please, find below the referees’ comments repeated and our responses inserted in after each comment.

REVIEWER 1.   COMMENTS TO AUTHOR:

As described above, I think the issue is very relevant.
However, I believe a moderate review of English language and form is essential. 

Response: We forwarded the document to an American doctor who was unable to locate anything that needed to be changed or corrected. He said the grammar and ortograph are both acceptable.

Reviewer 3 Report

I wish to compliment the reviewers with the improved manuscript. A few minor comments remain:

Table 1: An implanted cardiac device is a relative contra-indication for CMR. Patients with MRI compatible devices are suitable for CMR. Please specify. 

Line 405: (not sure about this phrase)?

Author Response

We would like to thank the 2  reviewers for their thoughtful review of the manuscript. They raise important issues and their inputs are very helpful for improving the manuscript. We agree with all their comments and we have revised our manuscript accordingly.

We marked with red color the modifications we have made in the revised manuscript.

Please, find below the referees’ comments repeated and our responses inserted in after each comment.

REVIEWER 3.   COMMENTS TO AUTHOR:

  • I wish to compliment the reviewers with the improved manuscript. A few minor comments remain:

Table 1: An implanted cardiac device is a relative contra-indication for CMR. Patients with MRI compatible devices are suitable for CMR. Please specify. 

Response: Thank you. We made a comment on MRI-safe devices.

  • Line 405: (not sure about this phrase)?

Response: Thank you. We eliminated the text.